# Prevalence estimates of putatively pathogenic leptin variants in the gnomAD database

**Luisa Sophie Rajcsanyi**[1,2], **Yiran Zheng**[1,2], **Pamela Fischer-Posovszky**[3], **Martin Wabitsch**[3], **Johannes Hebebrand**[1], **Anke Hinney**[1,2]*

**1** Department of Child and Adolescent Psychiatry, Psychosomatics and Psychotherapy, University Hospital Essen, University of Duisburg-Essen, Essen, Germany, **2** Center for Translational Neuro- and Behavioural Sciences, University Hospital Essen, Essen, Germany, **3** Department of Paediatrics and Adolescent Medicine, University Medical Center Ulm, Ulm, Germany

* anke.hinney@uni-due.de

## Abstract

Homozygosity for pathogenic variants in the leptin gene leads to congenital leptin deficiency causing severe early-onset obesity. This monogenic form of obesity has mainly been detected in patients from consanguineous families. Prevalence estimates for the general population using the Exome Aggregation Consortium (ExAC) database reported a low frequency of leptin mutations. One in approximately 15 million individuals will be homozygous for a deleterious leptin variant. With the present study, we aimed to extend these findings utilizing the augmented Genome Aggregation Database (gnomAD) v2.1.1 including more than 140,000 samples. In total, 68 non-synonymous and 7 loss-of-function leptin variants were deposited in gnomAD. By predicting functional implications with the help of *in silico* tools, like SIFT, PolyPhen2 and MutationTaster2021, the prevalence of hetero- and homozygosity for putatively pathogenic variants (n = 32; pathogenic prediction by at least two tools) in the leptin gene were calculated. Across all populations, the estimated prevalence for heterozygosity for functionally relevant variants was approximately 1:2,100 and 1:17,830,000 for homozygosity. This prevalence deviated between the individual populations. Accordingly, people from East Asia and individuals of mixed ethnicities ('Others') were at greater risk to carry a possibly damaging leptin variant. Generally, this study emphasises the scarcity of pathogenic leptin variants in the general population with varying prevalence for distinct study groups.

## Introduction

The leptin-melanocortin system modulates the energy homeostasis and body weight regulation via the hypothalamic arcuate nucleus (ARC). The hormone leptin (LEP) is secreted by the adipose tissue into the bloodstream. In the ARC, LEP binds to the leptin receptor on pro-opiomelanocortin (POMC) and agouti-related peptide (AgRP) expressing neurons, stimulating POMC's release and inhibiting AgRP's expression. Subsequently, POMC is post-translationally processed into the α-melanocyte-stimulating hormone. Eventually, the signalling of

obtained from the publicly available gnomAD database (https://gnomad.broadinstitute.org/).

**Funding:** This study was funded by the Deutsche Forschungsgemeinschaft (DFG; A.H.: HI 865/2-1; P.F.P.: Heisenberg professorship; project number: 398707781), the Bundesministerium für Bildung & Forschung (BMBF; A.H.: 01GS0820; PALGER 2017-33: 01DH19010). We further acknowledge support by the Open Access Publication Fund of the University of Duisburg-Essen. The funders had no role in study design, data collection and analysis, decision to publish or preparation of the manuscript.

**Competing interests:** The authors have declared that no competing interests exist.

melanocortin-4-receptor is stimulated leading to a decreased food intake due to satiety signals [1–5].

Homozygous mutations in the *LEP* gene cause congenital leptin deficiency disrupting the normal regulation of the body weight. Leptin levels in homozygous carriers of deleterious mutations are in most cases extremely low to undetectable [5, 6]. Some deleterious mutations lead to a biologically inactive leptin. Leptin levels in these patients are seemingly normal for their body mass index (BMI) [6]. A rapid weight gain eventually leads to extreme obesity with hyperphagia, hypogonadism and impaired immune functions being concomitant symptoms [5, 7–10]. This form of monogenic obesity is infrequent, with a prevalence between 1 and 5% and predominantly affecting individuals with parental consanguinity [5, 7, 8, 11–14]. In 1997, the first deleterious *LEP* mutation (p.Gly133Val*fs*\*15) was reported by Montague and colleagues [7]. It was detected in the homozygous state in two cousins descending from a consanguineous family with the unaffected parents being heterozygous for the variant. Due to this frameshift mutation, the LEP protein was truncated as 14 aberrant amino acids and a premature stop codon were introduced. This led to a rapid onset of obesity after normal birth weight [7]. Subsequent treatment with recombinant leptin led to a substantial weight loss and a decrease in energy intake [11, 15]. Further, besides frameshift mutations, pathogenic nonsense, and non-synonymous variants as well as deletions in *LEP* have been reported [5]. The functional effects of these mutations are diverse. For instance, a deletion (p.Ile35del) detected in two homozygous obese patient leads to a complete loss of the second exon of *LEP* and the removal of an isoleucine from the N-terminus of the protein [16, 17]. Additionally, the non-synonymous variant p.Asp100Tyr was detected in an extremely obese boy from a consanguineous family. He showed high serum leptin levels and a pronounced history of infections. Functional analyses revealed normal leptin expression and secretion but a dysfunctional bioinactive leptin that did not induce Stat3 phosphorylation [6, 14].

In 2017, Nunziata et al. [18] estimated the prevalence of putatively damaging mutations in the *LEP* gene using the Exome Aggregation Consortium (ExAC) database. Based on data from 60,706 samples, it was estimated that one in 15,000,000 individuals is potentially a homozygous carrier of a deleterious *LEP* mutation (determined by *in silico* tools), while approximately one in 2,000 individuals harbours a heterozygous leptin variant [18]. Upon inclusion of functionally relevant *LEP* variants described in the literature, the authors estimated higher prevalence of hetero- and homozygosity of 1:1,050 and 1:4,400,000, respectively [18]. To date, ExAC has been augmented into the Genome Aggregation Database (gnomAD) including more than 140,000 samples (version v2.1.1) [19]. Therefore, we aimed to estimate the prevalence of putatively deleterious non-synonymous, frameshift and nonsense (loss-of-function; LoF) mutations in the *LEP* gene based on this extended dataset represented in gnomAD v2.1.1.

## Materials and methods

### gnomAD

The gnomAD database (https://gnomad.broadinstitute.org/, accessed: Jan 24th, 2022) [19], encompasses 15,708 whole-genome and 125,748 exome sequencing datasets from individuals of various populations (v2.1.1, GRCh37/hg19) comprising more than 200 million genetic variants. The sequencing data predominantly originates from case-control studies of diseases diagnosed in adulthood, such as cardiovascular diseases or psychiatric disorders. To ensure high quality data, all samples were subjected to a quality control, excluding samples with low sequencing quality, samples from second-degree relatives or higher, and data from patients with severe paediatric diseases. In total, six global and eight sub-continental populations are included, while populations from the Middle East, Central and Southeast Asia, Oceania and

Africa being generally underrepresented. The mean coverage of the *LEP* gene was ~ 80x for exome and ~ 30x for genome data [19].

## Leptin variants and their predicted functional implications

In gnomAD, the *LEP* gene (canonical transcript ENST00000308868.4) was analysed and data pertaining to non-synonymous and LoF variants as well as the corresponding population-specific allele counts, and frequencies were extracted (see S1 Table).

Consequences on the leptin protein by non-synonymous variants were predicted utilizing various *in silico* tools, namely Sorting Intolerant From Tolerant (SIFT) [20], Polymorphism Phenotyping v2 (PolyPhen2) [21], MutationTaster2021 [22], Functional Analysis through Hidden Markov Models–multiple kernel learning (FATHMM-MKL) [23] and Protein Variation Effect Analyzer (PROVEAN) [24]. Predictions by SIFT, FATHMM-MKL and PROVEAN were obtained with the help of the Variant Effect Predictor (VEP) [25]. For LoF variants, gnomAD presents predictions whether the respective LoF variant is a high- or low-confidence LoF based on results of either the LOFTEE tool or a manual curation, shown below the information of VEP on gnomAD's variant page [19, 26]. SIFT classifies variants as either 'tolerated' or 'deleterious', while PolyPhen2 categorizes the mutations into 'benign', 'possibly damaging' and 'probably damaging'. For PolyPhen2, the 'HumVar' classifier model was applied. MutationTaster2021 subjects each variant to several *in silico* tools itself and subsequently annotates each substitution as either 'benign' or 'deleterious'. FATHMM-MKL and PROVEAN classify the variants into two categories: 'neutral' and 'damaging'. Except MutationTaster2021, all these tools exclusively analyse non-synonymous variants. Thus, we annotated frameshift mutations with the LoF confidence predictions stated on the variant's page.

To obtain additional hints for a putative clinical significance of a given variant (non-synonymous and LoF), the database ClinVar (https://www.ncbi.nlm.nih.gov/clinvar/) [27] was checked.

Based on the preceding *in silico* analyses, the probability of hetero- and homozygous variants predicted to be pathogenic was calculated applying the Hardy-Weinberg equilibrium (HWE) with the assumption of a perfect population (see Eq (1); p = allele frequency of allele A, q = allele frequency of allele a) as performed by Nunziata et al. [18].

$$p^2 + 2pq + q^2 = 1 \tag{1}$$

Hence, the prevalence of the heterozygous (*2qp*) and homozygous (including compound heterozygous; $q^2$) variants were determined (see Eq (1)). To assess the prevalence of homozygous variants, the frequencies ($q^2$) of the individual alleles were calculated and subsequently summed up. Subtraction of the prevalence of homozygosity from the prevalence of homozygous including the compound heterozygous variants revealed the corresponding frequencies for the compound heterozygotes.

When analysing the individual populations, substitutions were considered pathogenic if at least two of the applied *in silico* tools identified the variant as 'damaging' or 'deleterious' or if it was a high-confidence LoF variant.

Further, a literature search was performed. The PubMed database was screened for the term 'congenital leptin deficiency' and each individual non-synonymous or LoF variant extracted from gnomAD, to compile a list containing all obese subjects carrying a *LEP* variant and putative functional implications. This list was extended with references for each individual variant deposited in NCBI (https://www.ncbi.nlm.nih.gov/), Online Mendelian Inheritance in Man (OMIM; https://www.omim.org/), Ensembl (https://www.ensembl.org/) and LitVar (https://www.ncbi.nlm.nih.gov/CBBresearch/Lu/Demo/LitVar/) and two review articles [18, 28]. Allele counts were derived from gnomAD.

## Results

In total, 75 non-synonymous and LoF variants in the *LEP* gene were deposited in gnomAD. Of these, 68 were non-synonymous (90.70%), five were frameshift (6.67%) and one each was an in-frame deletion (1.33%) or splice acceptor variant (1.33%). Across all populations, the non-synonymous variant rs17151919 (p.Val94Met) was the most frequent polymorphism with an overall allele frequency (AF) of 0.84% (see Table 1 and S1 Table). A total of 105 homozygous and 2,167 heterozygous carriers of rs17151919 were observed (see S1 and S2 Tables). Yet, *in silico* tools predicted a non-pathogenic potential (see S2 Table).

Considering the populations individually, the samples of the group 'Others' for which no population could unambiguously be assigned, showed the highest occurrence of non-synonymous and LoF variants when correcting for the respective population size and assuming that all variants are equally frequent (0.0022; eight variants in total; see Table 1 and S1 Table). Occurrence rates of *LEP* variants in populations from East and South Asian countries were lower with 0.0014 (total of 14 variants) and 0.0010 (total of 16 variants), respectively. Within the African-American population, non-synonymous and LoF variants showed a population size-corrected frequency of 0.00104 (total of 13 variants). Lower occurrences were detected in the Latino/Admixed population (0.0008; total of 14 variants), the European, non-Finnish (0.0006; total of 41 variants), the European, Finnish (0.00024; total of three variants) and the Ashkenazi Jewish population (0.0002; one variant; see Table 1 and S1 Table).

Generally, for all populations, the majority of variants was annotated as non-synonymous mutations (> 85%) and were rare (AF < 1%, see Table 1 and S1 Table). Solely, the non-synonymous and putatively benign single nucleotide polymorphism (SNP) rs17151919 (see S1 and S2 Tables) was frequent in African-Americans with an AF of 8.4%. Further, this SNP was the most commonly detected variant in European, non-Finnish individuals as well as in the African-American, Latino/Admixed American, Ashkenazi Jewish, South Asian and 'Others' populations (see Table 1 and S1 Table). Conversely, in Finnish samples, the variant rs751272426 (AF = 0.0047%) was the most common, while rs148407750 (AF = 0.311%) was the most abundant variant in people from East Asia.

**Table 1. Summary of non-synonymous and LoF variants in the LEP gene deposited in gnomAD.**

| Population | Sample size | Total number of variants* | Number of non-synonymous variants | Number of LoF variants | Most common variant (AF) |
|---|---|---|---|---|---|
| All populations | 141,456 | 75 | 68 | 7 | rs17151919 (0.84%) |
| All populations (females) | 64,754 | 49 | 45 | 4 | rs17151919 (1%) |
| All populations (males) | 76,702 | 50 | 45 | 5 | rs17151919 (0.71%) |
| African-American | 12,487 | 13 | 12 | 1 | rs17151919 (8.41%) |
| Ashkenazi Jewish | 5,185 | 1 | 1 | 0 | rs17151919 (0.26%) |
| East-Asian | 9,977 | 14 | 13 | 1 | rs148407750 (0.31%) |
| European, Finnish | 12,562 | 3 | 3 | 0 | rs751272426 (0.04%) |
| European, non-Finnish | 64,603 | 41 | 37 | 4 | rs17151919 (0.04%) |
| Latino/Admixed American | 17,720 | 14 | 14 | 0 | rs17151919 (0.45%) |
| Others[a] | 3,614 | 8 | 8 | 0 | rs17151919 (0.38%) |
| South Asian | 15,308 | 16 | 14 | 2 | rs17151919 (0.03%) |

Table 1 summarizes the number of *LEP* variants (*non-synonymous and LoF) deposited in gnomAD (see S1 Table for full dataset). The most common variants detected in various populations were predicted to be benign (see S2 Table). The population with the term 'Others' refers to individuals of mixed population, for whom an unambiguous ethnicity could not be assigned ([a]). AF: allele frequency. LoF: loss-of-function.

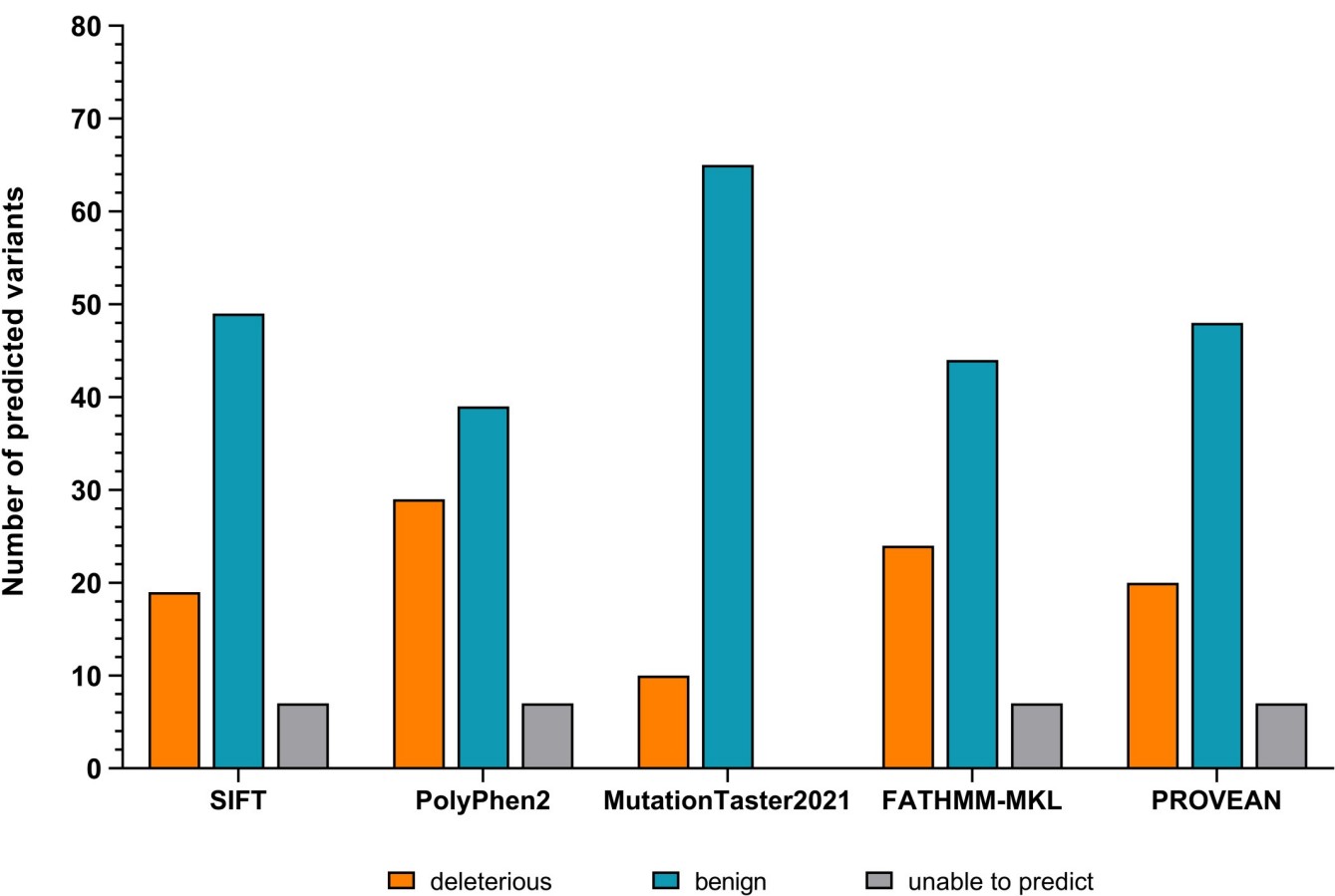

**Fig 1. Predictions of the applied in silico tools.** All 75 non-synonymous and LoF variants located in LEP were analysed with SIFT, PolyPhen2, MutationTaster2021, FATHMM-MKL and PROVEAN. Unless MutationTaster2021, all tools were unable to predict implications of the seven LoF variants (grey).

To assess functional implications of the *LEP* variants in gnomAD, we initially analysed the variants with various *in silico* tools (see S2 Table). Accordingly, SIFT assigned 19 variants as 'deleterious', while PolyPhen2 predicted 13 variants to be 'possibly damaging' and 16 to be 'probably damaging'. Ten variants were assigned as 'deleterious' by MutationTaster2021. 'Damaging' classifications for 24 and 20 variants were obtained by FATHMM-MKL and PRO-VEAN, respectively (see Fig 1 and S2 Table). Additionally, five of six LoF variants were indicated to be high-confidence LoF variants or in-frame deletions (see S2 Table).

In ClinVar, which was examined as an additional pathogenicity prediction tool, solely six of the 75 non-synonymous and LoF gnomAD variants were deposited (not shown). Of these, four were of 'uncertain significance', while the remaining two were predicted to be 'benign' (rs17151919 and rs28954113). The preceding *in silico* analyses have already assigned rs17151919 as 'benign', whereas rs28954113 was classified as 'deleterious' by all five computational tools. Further, previous studies have reported clinical cases with severe obesity caused by the amino acid exchange of rs28954113 (p.Asn103Lys) [6, 29–31]. Accordingly, we retained the classification of rs28954113 as 'pathogenic'.

Thus, twenty-two variants across all populations were predicted to be 'benign' (see Table 2 and S2 Table). Fifty-three variants were indicated to be 'pathogenic' by at least one *in silico* tool, while 32 and 19 revealed a pathogenic effect in at least two and three tools, respectively. Collectively, one in approximately 53 individuals will be a carrier of a non-synonymous or LoF

**Table 2. Estimated prevalence of hetero- and homozygous as well as compound heterozygous variants across all populations.**

| Number of tools predicting pathogenic effect | Number of (pathogenic) variants* | Number of carriers of pathogenic variants | | Estimated prevalence of heterozygous mutations | Estimated prevalence of homozygous and compound heterozygous mutations | Estimated prevalence of homozygous mutations | Estimated prevalence of compound heterozygous mutations |
|---|---|---|---|---|---|---|---|
| | | Heterozygous | Homozygous | | | | |
| 0 | 22[a, b] | 2,256[b] | 105[b] | 1: 58 | 1: 13,200 | 1: 14,200 | 1: 186,000 |
| ≥ 0 | 75[b] | 2,486[b] | 105[b] | 1: 53 | 1: 11,000 | 1: 14,100 | 1: 50,000 |
| ≥ 1 | 53 | 230 | 0 | 1: 616 | 1: 1,510,000 | 1: 7,050,000 | 1: 1,930,000 |
| ≥ 2 | 32 | 67 | 0 | 1: 2,100 | 1: 17,830,000 | 1: 328,200,000 | 1: 18,850,000 |
| ≥ 3 | 19 | 32 | 0 | 1: 4,400 | 1: 78,160,000 | 1: 741,000,000 | 1: 87,410,000 |

Here, the rounded estimated prevalence for variants across all populations applying different pathogenicity definitions are presented. Variants were considered deleterious if the stated number of *in silico* tools revealed a pathogenic prediction (*). If no tool indicated to a damaging effect ([a]), the variant is likely 'benign'. Due to varying allele frequencies across the individual populations, the Hardy-Weinberg equilibrium is not fulfilled when investigating all or exclusively benign variants ([b]).

variant located in *LEP* regardless of the pathogenicity (see Table 2). The prevalence for a homozygous and compound heterozygous variant was ~ 1:14,100 and ~ 1:50,000, respectively. Slightly lower prevalence were detected for variants predicted to be 'benign' (see Table 2 and S2 Table). Generally, for indicated pathogenic variants across all populations, if at least one tool predicted pathogenicity, the prevalence of compound heterozygosity is higher than the prevalence of homozygous variants (see Table 2).

Further, when applying various pathogenicity definitions based on the number of *in silico* tools predicting a damaging effect, it is evident that the more stringent this definition, the lower the prevalence (see Table 2). Consequently, we decided to classify variants as 'pathogenic' if at least two *in silico* tools indicated a deleterious impact (definition applied for subsequent analyses). A total of 67 individuals throughout all populations carried at least one of these variants heterozygously, while no homozygous carriers were detected. Hence, the estimated prevalence of heterozygosity for a putatively harmful *LEP* mutation was approximately 1:2,100, while the prevalence for a homozygous variant was ~ 1:17,830,000 for individuals of all populations (see Tables 2 and 3).

**Table 3. Estimated prevalence of putatively pathogenic LEP variants for the individual populations in gnomAD.**

| Population | Sample size | Number of putatively deleterious mutations* | Estimated prevalence for heterozygous mutations | Estimated prevalence for homozygous/compound heterozygous mutations |
|---|---|---|---|---|
| All populations | 141,456 | 32 | 1: 2,100 | 1: 17,830,000 |
| All populations (females) | 64,754 | 19 | 1: 2,200 | 1: 18,640,000 |
| All populations (males) | 76,702 | 22 | 1: 2,100 | 1: 17,190,000 |
| African-American | 12,487 | 4 | 1: 1,800 | 1: 12,730,000 |
| Ashkenazi Jewish | 5,185 | 0 | *NA* | *NA* |
| East-Asian | 9,977 | 6 | 1: 770 | 1: 2,360,000 |
| European, Finnish | 12,562 | 0 | *NA* | *NA* |
| European, non-Finnish | 64,603 | 16 | 1: 2,700 | 1: 28,980,000 |
| Latino/Admixed American | 17,720 | 5 | 1: 3,000 | 1: 34,890,000 |
| Others[a] | 3,614 | 5 | 1: 720 | 1: 2,090,000 |
| South Asian | 15,308 | 3 | 1: 1,300 | 1: 6,510,000 |

Estimated rounded prevalence for deleterious *LEP* variants are shown. Variants were considered deleterious if at least two *in silico* tools revealed a pathogenic prediction (*). Individuals of the 'Others' population could not be assigned unambiguously to one of the other ethnicities ([a]). NA: not available.

gnomAD further provides sex-specific allele counts for each variant (see Table 1). Thus, we replicated the probability estimations of possibly pathogenic variants (as defined above) for both sexes separately. This revealed that about one in 2,200 women carries a heterozygous and possibly harmful *LEP* variant. In males, the prevalence of a heterozygous variant was marginally higher with ~1:2,100. The chance to harbour a homozygous/compound heterozygous, pathogenic variant in females was estimated to be ~1:18,640,000. For males, this prevalence was again higher at ~1:17,190,000.

Next, we determined the likelihood of a putatively deleterious *LEP* variant in the distinct populations. As none of the variants detected in the Finnish and Ashkenazi Jewish population was predicted to have a pathogenic effect, we were unable to calculate the correlated prevalence (see Table 3 and S2 Table). Generally, pronounced variations between the global populations were observed (see Table 3). Individuals whose ethnicity could not be clearly assigned ('Others') were determined to be at highest risk to harbour a putatively pathogenic *LEP* variant either hetero- or homozygously (see Table 3). The second greatest risk for being a pathogenic *LEP* variant carrier was detected for individuals of the East Asian population. Conversely, the lowest risk for both hetero- and homozygous variants was found in the Latino/Admixed population (see Table 3).

In order to expand the pathogenicity predictions with reported clinical cases, we have performed a literature search (e.g. PubMed, OMIM, etc.) and have found 20 variants reported in at least one clinical case (see S3 Table). Of these, five were listed in the non-synonymous and LoF variants extracted from gnomAD. In turn, three of those, have already been assigned as 'pathogenic' by our *in silico* analyses (by at least two tools). Generally, all other variants reported in a clinical case were not available in gnomAD. When we considered the variants declared as 'pathogenic' by at least two *in silico* tools and variants reported in a clinical case for our estimates, we obtained higher prevalence for heterozygous (1: 1,300) as well as homozygous carriers (1: 6,380,000) across all populations (see S4 Table). Likewise, higher or similar prevalence were found when repeating this calculation for the individual populations. Once again, individuals whose ethnicity could not be unambiguously assigned ('Others') and East Asians were at higher risk of being a carrier of a putatively pathogenic leptin variant (see S4 Table).

Additionally, we have conducted a literature search for any functional implication of the variants. In total, seven non-synonymous and LoF variants listed in gnomAD were found to be functionally characterised by either a comprehensive computational analyses or by *in vitro* studies (see S2 Table). Of these, six were already assigned as 'pathogenic' by our *in silico* analyses. Solely, rs17151919 has been previously classified as 'benign' (by *in silico* tools and Clin-Var),but was reported to be functionally relevant [32]. Calculating the prevalence estimates for variants that have been characterized as 'pathogenic' by at least two tools and have a functional relevance revealed equally higher prevalence as the inclusion of variants found in clinical cases (see S4 Table). This was again valid for all populations. Resembling higher prevalence estimates were detected when variants from case reports as well as variants with a functional implication were added to the mutations predicted as 'pathogenic' by *in silico* tools (see S4 Table). Again, the prevalence rates for the individual ethnicities vary considerably. Statistically, one in six African-Americans carries a heterozygous and pathogenic *LEP* variant, while the risk for being a carrier in Finnish Europeans is lower (1: 1,400; see S4 Table).

## Discussion

Homozygous pathogenic mutations in the leptin gene lead to a deficiency of biologically active leptin and cause severe early-onset obesity [5, 7, 8, 11, 14]. Through the implementation of

reference databases, such as ExAC and gnomAD, prevalence assessments of potentially harmful variants in the general population have become feasible. Yet, solely one study has explored the prevalence of *LEP* variants in the general population using these reference datasets [18]. As of today, the gnomAD database is the largest publicly available repository with data of genetic variants [26]. More than 125,000 exome and 15,000 whole-genome sequence datasets are contained in gnomAD v2.1.1 [19]. Based on these datasets, it had been estimated that each individual carries approximately 200 coding variants with allele frequencies less than 0.1%. Despite the large sample size, gnomAD will lack on average 27 ± 13 novel coding mutations per exome based on the current number of samples included [26]. The data contained in gnomAD has been subjected to a stringent quality control excluding data of participants with known severe paediatric diseases or related individuals [19, 26]. Notably, due to this removal of samples with known paediatric diseases, potentially relevant and pathogenic variants with regard to early manifested obesity may have been omitted. Additionally, variation data regarding global cohorts are deposited in gnomAD. Still, non-Finnish European samples are overrepresented, while samples from the Middle East, Central Asia and Africa are generally underrepresented [19]. Since congenital leptin deficiency caused by mutations in leptin are more prevalent in patients from Pakistan and the Middle East [5, 7, 11, 16], there is a lack of data pertaining to deleterious leptin mutations in the general Middle Eastern population. It can be assumed that higher incidence of putatively harmful variants might be observed in these populations. Additionally, no individual-level phenotype data is available. Thus, it is unclear whether the datasets might be skewed for overweight or obese individuals, which is feasible considering the globally increasing prevalence of both [33].

Across all populations, we detected that approximately one in 2,100 carries a potentially deleterious (at least two *in silico* tools indicated a pathogenicity) heterozygous variant in *LEP*. In addition, the prevalence of a homozygous variant across all populations was about 1:17,830,000. Despite the larger sample size and a resultant greater number of variants in gnomAD, our results resemble the estimated prevalence based on the ExAC database reported by Nunziata and colleagues [18]. Further, when including variants reported in clinical cases and functional studies, we also obtained higher prevalence rates for hetero- and homozygosity and were thus able to confirm the previous findings [18].

Heterozygous variants were estimated to be more prevalent in the general public. Previously, these were predominantly detected in healthy unaffected individuals [5, 7]. Heterozygous carriers generally show lower BMI z-scores and lower body fat percentages than homozygous individuals [34]. For some variants in *LEP*, heterozygous carriers suffering from obesity have been reported [35–37]. Still, it can be assumed that heterozygous variants generally have an additive effect on the carrier's body weight. Across all populations, we report that these compound heterozygous variants are less prevalent than single heterozygous mutations but more frequent than homozygosity. In addition, we observed deviations in prevalence rates between populations. For example, individuals from East Asia and individuals, for whom no ethnicities could be determined ('Others'), showed a higher prevalence of both hetero- and homozygous mutations than other populations. Strong disparities were also evident at the SNP level. For instance, the polymorphism rs17151919 was generally infrequently detected. In African-Americans, however, it was a common variant (AF > 5%), which has already been reported for other study groups [32]. Further, an African-American-specific association of rs17151919 with lower leptin levels was reported. This SNP was also associated with a higher BMI in African children, but not in adults [32].

We are aware that *in silico* tools are no substitute for functional *in vitro* analyses. This is particularly evident for the deletion p.Ile35del, as neither gnomAD, nor most *in silico* tools do provide predictions of functional implications. Yet, it is known that this deletion causes the

loss of exon 2 of the *LEP* gene and thus a congenital leptin deficiency with resultant obesity [5, 16]. Additionally, the performance of the individual tools varies considerably, even across different populations and variant types [38, 39]. For instance, SIFT and the predecessor of Poly-Phen2, PolyPhen, were found to perform better when predicting LoF than gain-of-function variants [38]. Likewise, the pathogenicity of variants with an AF < 1% across all populations or variants with an AF between 1 and 25% in individual ethnicities was shown to be more challenging to accurately predict [39]. Previously, one study has demonstrated that SIFT and PRO-VEAN yield the most accurate prediction of pathogenicity, while MutationTaster2021 and FATHMM had comparatively low accuracy and specificity [40]. Conversely, other studies have shown that especially SIFT, PolyPhen2 and MutationTaster2021 exhibited a high sensitivity but a low specificity [39, 41]. Thus, the usage and evaluation of diverse tools appears to be essential. Initially, we have tested, how the number of tools indicating a pathogenic effect, affected our prevalence estimates (see Table 2). We have seen that the more stringent this criterion of pathogenicity was defined, the lower the obtained prevalence. Accordingly, we classified variants as potentially harmful if at least two tools indicated a damaging effect. Still, it remains uncertain whether these classifications can be corroborated by clinical and functional data.

Due to these considerations, we have additionally checked the ClinVar database to obtain additional pathogenicity indications and have performed a literature search to find reported clinical cases carrying *LEP* variants and to identify mutations that have been described to be functionally relevant (see S2 Table). Notably, ClinVar solely contained six of the 75 non-synonymous and LoF variants listed in gnomAD. The majority of those were of 'uncertain significance', while two were assigned as 'benign'. Interestingly, one was predicted to be 'pathogenic' by all here investigated computational tools. This pathogenic indication was even supported by multiple clinical cases of severe obesity (see S3 Table) [6, 29–31]. Hence, further research is urgently required to elucidate the unambiguous significance of many *LEP* variants for the phenotype of congenital leptin deficiency.

Similarly, the literature search screening e.g., PubMed, OMIM and LitVar, determined 20 *LEP* mutations in total that were at least detected in one obese individual. Again, solely five of those were included in the variant list extracted from gnomAD. The non-synonymous variants p.Asp100Asn (rs724159998) [42], p.Asn103Lys (rs28954113) [6, 29–31], the frameshift mutation p.Gly133Val*fs*Ter15 (rs1307773933) [7, 15–17, 43–45] and the in-frame deletion p.Ile35-del (rs747703977) [16, 17] were detected in extremely obese patients being homozygous carriers [5]. Of these, solely one mutation (rs1800564) deviated in its pathogenicity predictions. Likewise, in functional studies, seven variants were included in the gnomAD list. Again, the only variant showing deviating pathogenicity classification between the *in silico* analyses and the literature, was rs17151919. As we have detected higher prevalence rates when including variants reported either in clinical cases or functional studies, these might be caused by the allele counts and frequencies of rs17151919 and rs1800564. These two variants were more frequently found than the rest of the non-synonymous and LoF variants in gnomAD. For instance, relative to the majority of the as 'pathogenic' predicted variants by the *in silico* tools, rs1800564 has been found more frequently in the gnomAD population (43 heterozygous carriers) and thus presumably accounts for the higher prevalence rates. The same applies to the SNP rs17151919.

Despite all these remarks that need to be considered in the interpretation of our results, *in silico* tools do help to gain preliminary indications of putatively pathogenic variants. It is even recommended by the American College of Medical Genetics and Genomics (ACMG) and the European Society of Human Genetics (ESHG) to use computational predictions to support the interpretation of variants [39].

Generally, the application of the HWE is affected by several factors, like mutations, natural selection, non-random mating, genetic drift, gene flow, population structures and sizes [46, 47]. For instance, for the fulfilment of the HWE an infinite population size is assumed. Yet, this can never be met by any population in nature [47]. Further, the 'Wahlund effect' influences the HWE. In populations with multiple subpopulations, individuals might mate within those subpopulations but never between them, resulting in an underestimation of homozygotes by the HWE in the overall population [46]. Generally, it is challenging to predict the impact of the gnomAD populations and their characteristics on the HWE and thus our results.

## Conclusion

The gnomAD database is the largest publicly available reference dataset including various global study groups. By utilizing these datasets, we estimated the prevalence of putatively damaging leptin variants. We identified 32 possibly damaging mutations in 67 heterozygous and no homozygous carriers. The estimated prevalence of a heterozygous variant was roughly 1:2,100, while the probability for homozygosity was 1:17,830,000 across all populations. Investigating each study group separately, this prevalence varied significantly, with individuals of mixed, unknown ethnicity ('Others') and East Asians being at greater risk of harbouring a hetero- or homozygous mutation with a harmful consequence. Yet, higher prevalence of functionally relevant variants were obtained upon inclusion of reported case and functional studies. In general, mutations in the *LEP* gene, which frequently result in congenital leptin deficiency, are extremely rare in the general population. Continued analysis of leptin mutations along phenotypic and clinical data may improve our understanding of monogenic obesity.

## Supporting information

**S1 Table. Raw data as extracted from gnomAD.** This contains the raw data as it was extracted from gnomAD [19]. This includes the 75 non-synonymous and LoF *LEP* variants with its corresponding allele counts and frequencies. It represents the minimal dataset underlying the performed analyses.
(XLSX)

**S2 Table. Summary of the results of the *in silico* analyses of *LEP* variants deposited in gnomAD.** This represents the *in silico* predictions for each variant present in all populations by various tools, namely SIFT [20], PROVEAN [24], PolyPhen2 [21], MutationTaster2021 [22] and FATHMM-MKL [23]. The tools are ordered by their reported accuracy (left: highest accuracy; right: lowest accuracy) [40]. The literature references refer either to a reported clinical case or a functional study.
(PDF)

**S3 Table. Reported cases with *LEP* mutations.** This represents all clinical cases reported with either obesity and/or congenital leptin deficiency carrying a *LEP* variant. Not all reported variants were listed in gnomAD. NA: not available.
(PDF)

**S4 Table. Estimated prevalence for the populations in gnomAD based on *in silico* tools and literature references.** This presents the prevalence estimations using varying definitions of pathogenicity. Here, all variants are classified as 'pathogenic' if they were predicted as harmful by at least two *in silico* tools and were either reported in a clinical case or in a functional study. Further, solely variants listed in the non-synonymous and LoF *LEP* variants of gnomAD were included in the calculations. NA: not available.
(PDF)

## Author Contributions

**Conceptualization:** Luisa Sophie Rajcsanyi, Yiran Zheng, Pamela Fischer-Posovszky, Martin Wabitsch, Johannes Hebebrand, Anke Hinney.

**Data curation:** Luisa Sophie Rajcsanyi.

**Formal analysis:** Luisa Sophie Rajcsanyi, Yiran Zheng.

**Investigation:** Luisa Sophie Rajcsanyi.

**Methodology:** Luisa Sophie Rajcsanyi, Yiran Zheng, Pamela Fischer-Posovszky, Martin Wabitsch, Johannes Hebebrand, Anke Hinney.

**Project administration:** Luisa Sophie Rajcsanyi, Pamela Fischer-Posovszky, Martin Wabitsch, Johannes Hebebrand, Anke Hinney.

**Supervision:** Pamela Fischer-Posovszky, Martin Wabitsch, Johannes Hebebrand, Anke Hinney.

**Validation:** Yiran Zheng.

**Visualization:** Luisa Sophie Rajcsanyi.

**Writing – original draft:** Luisa Sophie Rajcsanyi.

**Writing – review & editing:** Yiran Zheng, Pamela Fischer-Posovszky, Martin Wabitsch, Johannes Hebebrand, Anke Hinney.

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
