## [Decision Letter · Decision Letter 0]

12 May 2022

PONE-D-22-08349

Prevalence Estimates of Putatively Pathogenic Leptin Variants in the gnomAD Database

PLOS ONE

Dear Dr. %Hinney%,

Thank you for submitting your manuscript to PLOS ONE. After careful consideration, we have decided that your manuscript does not meet our criteria for publication and must therefore be rejected.

Specifically: This study does not provide any novel information to the scientific world as mentioned by one of the reviewer. However I am sure comments of the learned reviewer will help you to do a better analysis at not only a gene level but at genome or disease level.

I am sorry that we cannot be more positive on this occasion, but hope that you appreciate the reasons for this decision.

Kind regards,

Tiratha Raj Singh

Academic Editor

PLOS ONE

Reviewers' comments:

Reviewer's Responses to Questions

**Comments to the Author**

1. Is the manuscript technically sound, and do the data support the conclusions?

Reviewer #1: Yes

2. Has the statistical analysis been performed appropriately and rigorously? 

Reviewer #1: N/A

3. Have the authors made all data underlying the findings in their manuscript fully available?

Reviewer #1: Yes

4. Is the manuscript presented in an intelligible fashion and written in standard English?

Reviewer #1: Yes

5. Review Comments to the Author

Reviewer #1: Dear authors,

It was my pleasure to read your manuscript entitled “Prevalence Estimates of Putatively Pathogenic Leptin Variants in the gnomAD Database”, but unfortunately I am suggesting the journal to reject it for publication.

My recommendation is based on the fact that it doesn’t add anything new and the whole paper is actually based on a very simple calculation. How many potentially pathogenic variants have been identified in the investigated gene, extract the combined minor allele frequency and then by using the Hardy Weinberg equilibrium, to estimate the prevalence of affected individuals.

As I hope you can understand, this calculation while valid (I would suggest though using the ACMG criteria and not merely basing potential pathogenicity on a majority vote among prediction algorithms) and well presented, is not enough for a full research article. If you are interested in pursuing further this type of research I would suggest to concentrate not on one gene, but a disease, correlate with clinical data and draw conclusions about the concordance or discordance between them, while making a mini literature review of the subject.

6. PLOS authors have the option to publish the peer review history of their article (what does this mean?). If published, this will include your full peer review and any attached files.

Reviewer #1: No

- - - - -

---

## [Author Response · Author response to Decision Letter 0]

31 May 2022

Comments to the Author

1. Is the manuscript technically sound, and do the data support the conclusions?

Reviewer #1: Yes

Response: We thank the reviewer for the positive evaluation.

2. Has the statistical analysis been performed appropriately and rigorously?

Reviewer #1: N/A

Response: We have performed the (statistical) analysis appropriately and rigorously. We have even added more analyses and data regarding the leptin gene as a comparable study very recently published (PLoS One) for the FTO gene in weight regulation (Souza Junior et al. 2022 PMID: 34990463).

3. Have the authors made all data underlying the findings in their manuscript fully available?

Reviewer #1: Yes

Response: We again thank the reviewer for the positive evaluation.

4. Is the manuscript presented in an intelligible fashion and written in standard English?

Reviewer #1: Yes

Response: We thank the reviewer. 

5. Review Comments to the Author

Reviewer #1: Dear authors,

It was my pleasure to read your manuscript entitled "Prevalence Estimates of Putatively Pathogenic Leptin Variants in the gnomAD Database", but unfortunately I am suggesting the journal to reject it for publication.

My recommendation is based on the fact that it doesn't add anything new and the whole paper is actually based on a very simple calculation. How many potentially pathogenic variants have been identified in the investigated gene, extract the combined minor allele frequency and then by using the Hardy Weinberg equilibrium, to estimate the prevalence of affected individuals.

As I hope you can understand, this calculation while valid (I would suggest though using the ACMG criteria and not merely basing potential pathogenicity on a majority vote among prediction algorithms) and well presented, is not enough for a full research article. If you are interested in pursuing further this type of research I would suggest to concentrate not on one gene, but a disease, correlate with clinical data and draw conclusions about the concordance or discordance between them, while making a mini literature review of the subject.

Response: 

We are grateful for the very positive evaluation and that the manuscript was a pleasurable read. Thus, we hope that other reader would also profit from our manuscript. 

We like to point out our novel findings. Our analysis adds to the scientific knowledge pertaining to the leptin gene, as we have calculated not only the prevalence of homozygotes and heterozygotes of leptin variants, but also of carriers of compound heterozygous variants. This has never been reported before for leptin variants derived from a public database. We are the first to report frequencies derived from the currently largest database comprising more the 125,000 individuals (Gnomad). We have also calculated population-specific and gender-specific prevalences. These analyses have not been published before. 

We have also checked a large number of in silico tools to obtain educated knowledge about the functional relevance or pathogenicity of the detected variants.

Thus, our study adds substantial knowledge to genetic variations in the leptin gene and the implications for obesity in general. 

To point out that the HWE test is appropriate (as pointed out by the reviewer) we added a respective reference (Lines 128-129): ‘assumption of a perfect population (see Equation (1); p = allele frequency of allele A, q = allele frequency of allele a) as performed by Nunziata et al. (17).

---

## [Decision Letter · Decision Letter 1]

12 Jul 2022

PONE-D-22-08349R1Prevalence Estimates of Putatively Pathogenic Leptin Variants in the gnomAD DatabasePLOS ONE

Dear Dr. Hinney,

Thank you for submitting your manuscript to PLOS ONE. After careful consideration, we feel that it has merit but does not fully meet PLOS ONE’s publication criteria as it currently stands. Therefore, we invite you to submit a revised version of the manuscript that addresses the points raised during the review process.

We look forward to receiving your revised manuscript.

Kind regards,

Alvaro Galli

Academic Editor

PLOS ONE

Journal Requirements:

3. Please expand the acronym “BMBF” (as indicated in your financial disclosure) so that it states the name of your funders in full.

5. Please upload a new copy of Figure 1 as the detail is not clear. Please follow the link for more information: https://blogs.plos.org/plos/2019/06/looking-good-tips-for-creating-your-plos-figures-graphics/" https://blogs.plos.org/plos/2019/06/looking-good-tips-for-creating-your-plos-figures-graphics/

Additional Editor Comments (if provided):

Reviewers' comments:

Reviewer's Responses to Questions

**Comments to the Author**

1. If the authors have adequately addressed your comments raised in a previous round of review and you feel that this manuscript is now acceptable for publication, you may indicate that here to bypass the “Comments to the Author” section, enter your conflict of interest statement in the “Confidential to Editor” section, and submit your "Accept" recommendation.

Reviewer #2: (No Response)

Reviewer #3: (No Response)

Reviewer #4: (No Response)

2. Is the manuscript technically sound, and do the data support the conclusions?

Reviewer #2: Yes

Reviewer #3: Partly

Reviewer #4: Partly

3. Has the statistical analysis been performed appropriately and rigorously? 

Reviewer #2: N/A

Reviewer #3: N/A

Reviewer #4: Yes

4. Have the authors made all data underlying the findings in their manuscript fully available?

Reviewer #2: Yes

Reviewer #3: Yes

Reviewer #4: Yes

5. Is the manuscript presented in an intelligible fashion and written in standard English?

Reviewer #2: Yes

Reviewer #3: Yes

Reviewer #4: Yes

6. Review Comments to the Author

Reviewer #2: Introduction: Appropriate

Methodology: In accordance to goal of study, Used relative tools and equations. Tools used for various analysis are standard and have been used in similar studies.

Statistical analysis: Statistics applied is as per study demand

Results: Presented in comprehensive way and highlight the aim of study

Discussion: Could add more discussion on the methodology and tools used for prevalence estimations, this will add more weight-age and authenticity to results

Overall: Study design is good. Methodology is appropriate. Results are in accordance with previous reported individual studies. Discussion might need more substance

Reviewer #3: The authors submitted the prevalence estimates for homozygous/compound heterozygous LEP pathogenic variants. They estimated the pathogenicity of the variants registered to gnomAD from the results of several in silico tools' analysis results, and calculated the prevalence from the allele frequencies registered in the gnomAD based on the estimated pathogenicity. I strongly disagree to estimate the pathogenicity of variants only by in silico tools. At least, clinical information is needed. This kind of studies, we usually pick up only variants which were evaluated pathogenic by clinical databases such as ClinVar or HGMD professional. Those databases evaluate the pathogenicity by their own algorithms including clinical information. When we have clinical data, we can evaluate the pathogenicity by ourselves using ACMG criteria. However, this study lacks the evaluation of clinical information. I don't think this algorithm can determine the pathogenicity of varints in LEP.

Reviewer #4: Thank you for the invitation to review and sorry for the delay.

I have a few comments:

1. I think that through a systematic review the authors could derived greater confidence in which variants they are assessing as pathogenic or not. The phenotype of congenital leptin deficiency is very clear and with the widespread adpotion of exome sequencing for suspected monogenic obesity, the majority of pathogenic variants have probably been reported. In addition, there is extensive literature of in vitro functional characterisation of leptin mutants. The authors only have to consider a comparatively small number of variants (i.e. <100) so it would be possible to annotate each variant as to whether it had ever been described in a clinical case and whether it had been shown to be a loss of function (or hypomorphic) variant in vitro.

2. The leptin gene is highly resistant to loss of function variants and this analysis assumes that compound heterozygotes will be present at the expected rate based on the prevalence of the individual alleles. This may be true but I'm not sure we have evidence to support that. The public availability of the UK BioBank data would allow for identification of exactly the number of compound heterozygotes in a large cohort, though I appreciate this is a substantial undertaking compared to your current analysis. [It would also be biased by my point (3) below.]

3. The gnomAD database is primarily composed of individuals not known to have a monogenic disease. Leptin deficiency causes quite a dramatic phenotype of hyperphagia. I anticipate they they may be under-represented in gnomAD compared to their true prevalence. (Unlike for asymptomatic or very late-onset conditions.)

7. PLOS authors have the option to publish the peer review history of their article (what does this mean?). If published, this will include your full peer review and any attached files.

Reviewer #2: No

Reviewer #3: No

Reviewer #4: **Yes: **

---

## [Author Response · Author response to Decision Letter 1]

4 Aug 2022

Response to Editor

- The acronym ‘BMBF’ stands for ‘Bundesministerium für Bildung & Forschung‘.

- Our ‚financial disclosure‘ shall be as follows: ‘This study was funded by the Deutsche Forschungsgemeinschaft (DFG; A.H.: HI 865/2-1; P.F.P.: Heisenberg professorship; project number: 398707781), the Bundesministerium für Bildung & Forschung (BMBF; A.H.: 01GS0820; PALGER 2017-33: 01DH19010). We further acknowledge support by the Open Access Publication Fund of the University of Duisburg-Essen. The funders had no role in study design, data collection and analysis, decision to publish or preparation of the manuscript.’

- We have further added our minimal underlying dataset as our new S1 Table.

- We have updated the manuscript regarding to your recommendations.

Response to Reviewers

Response: We thank all Reviewers and the Academic Editor for the kind feedback to our manuscript “Prevalence estimates of putatively pathogenic leptin variants in the gnomAD database”! We hope that future readers will enjoy our work as well.

Generally, we have corrected some grammatical errors. By incorporating the feedback below, we have done some restructuring; particularly of the discussion. As we have cross- checked all our calculations, we have discovered some minor errors (especially rounding errors). These were also corrected.

Please find below our answers to the helpful comments (in italic). Our changes as found in the ms have been highlighted in red (see also Track Changes version). The lines stated are based on the Track Changes version (with a complete markup).

Reviewer #2:

Introduction: Appropriate

Methodology: In accordance to goal of study, Used relative tools and equations. Tools used for various analysis are standard and have been used in similar studies.

Statistical analysis: Statistics applied is as per study demand

Results: Presented in comprehensive way and highlight the aim of study

Response: We thank the Reviewer for this positive feedback.

Discussion: Could add more discussion on the methodology and tools used for prevalence estimations, this will add more weight-age and authenticity to results

Response: We thank the Reviewer for this comment. We have added a deeper discussion regarding the applied in silico tools, as well as a short paragraph regarding factors influencing the Hardy-Weinberg-Equilibrium. Please find our changes in lines 341 ff., 386 ff. and 423 ff.

Lines 341 ff.: Additionally, the performance of the individual tools varies considerably, even across different populations and variant types [38, 39]. For instance, SIFT and the predecessor of PolyPhen2, PolyPhen, were found to perform better when predicting LoF than gain-of-function variants [38]. Likewise, the pathogenicity of variants with an AF < 1% across all populations or variants with an AF between 1 and 25 % in individual ethnicities was shown to be more challenging to accurately predict [39]. Previously, one study has demonstrated that SIFT and PROVEAN yield the most accurate prediction of pathogenicity, while MutationTaster2021 and FATHMM had comparatively low accuracy and specificity [40]. Conversely, other studies have shown that especially SIFT, PolyPhen2 and MutationTaster2021 exhibited a high sensitivity but a low specificity [39, 41]. Thus, the usage and evaluation of diverse tools appears to be essential. Initially, we have tested, how the number of tools indicating a pathogenic effect, affected our prevalence estimates (see Table 2). We have seen that the more stringent this criteria of pathogenicity was defined, the lower the obtained prevalence. Accordingly, we classified variants as potentially harmful if at least two tools indicated a damaging effect. Still, it remains uncertain whether these classifications can be corroborated by clinical and functional data.

Lines 386 ff.: Despite all these remarks that need to be considered in the interpretation of our results, in silico tools do help to gain preliminary indications of putatively pathogenic variants. It is even recommended by the American College of Medical Genetics and Genomics (ACMG) and the European Society of Human Genetics (ESHG) to use computational predictions to support the interpretation of variants [39].

Lines 423 ff.: Generally, the application of the HWE is affected by several factors, like mutations, natural selection, non-random mating, genetic drift, gene flow, population structures and sizes [46, 47]. For instance, for the fulfilment of the HWE an infinite population size is assumed. Yet, this can never be met by any population in nature [47]. Further, the ‘Wahlund effect’ influences the HWE. In populations with multiple subpopulations, individuals might mate within those subpopulations but never between them, resulting in an underestimation of homozygotes by the HWE in the overall population [46]. Generally, it is challenging to predict the impact of the gnomAD populations and their characteristics on the HWE and thus our results.

Overall: Study design is good. Methodology is appropriate. Results are in accordance with previous reported individual studies. Discussion might need more substance

Response: We again thank the Reviewer for this positive feedback. We are grateful that our work and its relevance was recognised.

Reviewer #3:

The authors submitted the prevalence estimates for homozygous/compound heterozygous LEP pathogenic variants. They estimated the pathogenicity of the variants registered to gnomAD from the results of several in silico tools' analysis results, and calculated the prevalence from the allele frequencies registered in the gnomAD based on the estimated pathogenicity. I strongly disagree to estimate the pathogenicity of variants only by in silico tools. At least, clinical information is needed. This kind of studies, we usually pick up only variants which were evaluated pathogenic by clinical databases such as ClinVar or HGMD professional. Those databases evaluate the pathogenicity by their own algorithms including clinical information. When we have clinical data, we can evaluate the pathogenicity by ourselves using ACMG criteria. However, this study lacks the evaluation of clinical information. I don't think this algorithm can determine the pathogenicity of varints in LEP.

Response: We thank the Reviewer for this relevant critic. We are aware that in silico tools are no substitute to in vitro studies and studying these without clinical data might have some disadvantages. Still, we think that they give a reasonable first indication towards pathogenicity. To consider your comment, we have checked ClinVar and HGMD. Unfortunately, we do not have access to HGMD Professional. Yet, as another Reviewer suggested to perform a literature search to obtain information about all clinical data, we have already obtained all (and even additional) data deposited in the normal version of HGMD. In ClinVar, solely six of our 75 investigated variants of gnomAD were included. Of those, four were assigned to be of ‘uncertain significance’, while the remaining two (rs17151919 and rs28954113) were classified as ‘benign’. Yet, rs28954113 (p.Asn103Lys) is a variant that was reported in various clinical cases (see S3 Table) and was predicted to be ‘pathogenic’ by all five of our in silico tools. Thus, we retained the ‘pathogenic’ implication for this variant.

Therefore, the ClinVar analysis could not add further information about the pathogenicity of the variants. We still included our ClinVar examination in the ms, while we did not include the information regarding HGMD. Please find the updated information in lines 130-131, 196 ff. and 358 ff.

Lines 130-131: To obtain additional hints for a putative clinical significance of a given variant (non-synonymous and LoF), the database ClinVar (https://www.ncbi.nlm.nih.gov/clinvar/) [27] was checked.

Lines 196 ff.: In ClinVar, which was examined as an additional pathogenicity prediction tool, solely six of the 75 non-synonymous and LoF gnomAD variants were deposited (not shown). Of these, four were of ‘uncertain significance’, while the remaining two were predicted to be ‘benign’ (rs17151919 and rs28954113). The preceding in silico analyses have already assigned rs17151919 as ‘benign’, whereas rs28954113 was classified as ‘deleterious’ by all five computational tools. Further, previous studies have reported clinical cases with severe obesity caused by the amino acid exchange of rs28954113 (p.Asn103Lys) [6, 29-31].

Accordingly, we retained the classification of rs28954113 as ‘pathogenic’.

Lines 358 ff.: Due to these considerations, we have additionally checked the ClinVar database to obtain additional pathogenicity indications and have performed a literature search to find reported clinical cases carrying LEP variants and to identify mutations that have been described to be functionally relevant (see S2 Table). Notably, ClinVar solely contained six of the 75 non-synonymous and LoF variants listed in gnomAD. The majority of those were of ‘uncertain significance’, while two were assigned as ‘benign’. Interestingly, one was predicted to be ‘pathogenic’ by all here investigated computational tools. This pathogenic indication was even supported by multiple clinical cases of severe obesity (see S3 Table) [6, 29-31]. Hence, further research is urgently required to elucidate the unambiguous significance of many LEP variants for the phenotype of congenital leptin deficiency.

Reviewer #4:

Thank you for the invitation to review and sorry for the delay. I have a few comments:

1. I think that through a systematic review the authors could derived greater confidence in which variants they are assessing as pathogenic or not. The phenotype of congenital leptin deficiency is very clear and with the widespread adpotion of exome sequencing for suspected monogenic obesity, the majority of pathogenic variants have probably been reported. In addition, there is extensive literature of in vitro functional characterisation of leptin mutants. The authors only have to consider a comparatively small number of variants (i.e. <100) so it would be possible to annotate each variant as to whether it had ever been described in a clinical case and whether it had been shown to be a loss of function (or hypomorphic) variant in vitro.

Response: We thank the Reviewer for this valid comment. We have thus performed a literature search in PubMed (for ‘congenital leptin deficiency’ and each individual variant), OMIM (‘leptin deficiency’), NCBI (each variant), Ensembl (each variant) and LitVar (each variant) to obtain information about reported clinical cases and functional implications. The determined variants can be found in S3 Table. We have further annotated each variant whether it was reported in a clinical case or functional study in our S2 Table. If the resultant variants were included in the gnomAD non-synonymous and LoF variant list, we have re- calculated the prevalence estimates when including variants of clinical cases and functional studies. The results are represented in S4 Table. Please find further information in lines 146 ff., 258 ff. and 368 ff.

Lines 146 ff.: Further, a literature search was performed. The PubMed database was screened for the term ‘congenital leptin deficiency’ and each individual non-synonymous or LoF variant extracted from gnomAD, to compile a list containing all obese subjects carrying a

LEP variant and putative functional implications. This list was extended with references for each individual variant deposited in NCBI (https://www.ncbi.nlm.nih.gov/), Online Mendelian Inheritance in Man (OMIM; https://www.omim.org/), Ensembl (https://www.ensembl.org/) and LitVar (https://www.ncbi.nlm.nih.gov/CBBresearch/Lu/Demo/LitVar/) and two review articles [18, 28]. Allele counts were derived from gnomAD.

Lines 258 ff: In order to expand the pathogenicity predictions with reported clinical cases, we have performed a literature search (e.g. PubMed, OMIM, etc.) and have found 20 variants reported in at least one clinical case (see S3 Table). Of these, five were listed in the non- synonymous and LoF variants extracted from gnomAD. In turn, three of those, have already been assigned as ‘pathogenic’ by our in silico analyses (by at least two tools). Generally, all other variants reported in a clinical case were not available in gnomAD. When we considered the variants declared as ‘pathogenic’ by at least two in silico tools and variants reported in a clinical case for our estimates, we obtained higher prevalence for heterozygous (1 : 1,300) as well as homozygous carriers (1 : 6,380,000) across all populations (see S4 Table). Likewise, higher or similar prevalence were found when repeating this calculation for the individual populations. Once again, individuals whose ethnicity could not be unambiguously assigned ('Others') and East Asians were at higher risk of being a carrier of a putatively pathogenic leptin variant (see S4 Table).

Additionally, we have conducted a literature search for any functional implication of the variants. In total, seven non-synonymous and LoF variants listed in gnomAD were found to be functionally characterised by either a comprehensive computational analyses or by in vitro studies (see S2 Table). Of these, six were already assigned as ‘pathogenic’ by our in silico analyses. Solely, rs17151919 has been previously classified as ‘benign’ (by in silico tools and ClinVar), but was reported to be functionally relevant [32]. Calculating the prevalence estimates for variants that have been characterized as 'pathogenic' by at least two tools and have a functional relevance revealed equally higher prevalence as the inclusion of variants found in clinical cases (see S4 Table). This was again valid for all populations. Resembling higher prevalence estimates were detected when variants from case reports as well as variants with a functional implication were added to the mutations predicted as ‘pathogenic’ by in silico tools (see S4 Table). Again, the prevalence rates for the individual ethnicities vary considerably. Statistically, one in six African-Americans carries a heterozygous and pathogenic LEP variant, while the risk for being a carrier in Finnish Europeans is lower (1: 1,400; see S4 Table).

Lines 368 ff.: Similarly, the literature search screening e.g. PubMed, OMIM and LitVar determined 20 LEP mutations in total that were at least detected in one obese individual. Again, solely five of those were included in the variant list extracted from gnomAD. The non- synonymous variants p.Asp100Asn (rs724159998) [42], p.Asn103Lys (rs28954113) [6, 29-

31], the frameshift mutation p.Gly133ValfsTer15 (rs1307773933) [7, 15-17, 43-45] and the in- frame deletion p.Ile35del (rs747703977) [16, 17] were detected in extremely obese patients being homozygous carriers [5]. Of these, solely one mutation (rs1800564) deviated in its pathogenicity predictions. Likewise, in functional studies, seven variants were included in the gnomAD list. Again, the only variant showing deviating pathogenicity classification between the in silico analyses and the literature, was rs17151919. As we have detected higher prevalence rates when including variants reported either in clinical cases or functional studies, these might be caused by the allele counts and frequencies of rs17151919 and rs1800564. These two variants were more frequently found than the rest of the non- synonymous and LoF variants in gnomAD. For instance, relative to the majority of the as ‘pathogenic’ predicted variants by the in silico tools (highest number of heterozygous carriers in all populations was 7 for rs1307773933), rs1800564 has been found more frequently in the gnomAD population (43 heterozygous carriers) and thus presumably accounts for the higher prevalence rates. The same applies to the SNP rs17151919.

2. The leptin gene is highly resistant to loss of function variants and this analysis assumes that compound heterozygotes will be present at the expected rate based on the prevalence of the individual alleles. This may be true but I'm not sure we have evidence to support that. The public availability of the UK BioBank data would allow for identification of exactly the number of compound heterozygotes in a large cohort, though I appreciate this is a substantial undertaking compared to your current analysis. [It would also be biased by my point (3) below.]

Response: We again thank the Reviewer for this important feedback. Unfortunately, we do not have access to the UK Biobank data and because of the considerable time and, above all, costs incurred; it is beyond our capacity to gain access to the data. However, we might consider this approach for an individual paper in the future. But to still consider your comment, we analysed genotyping data from the NCBI ALFA (over 2 million subjects included) and 1000Genomes projects. Sadly, solely few of our analysed variants were included in these datasets. For instance, in the 1000Genomes project (Phase 3), deposited in Ensembl, only seven of the 75 variants were detected. Thus, we decided to not include these findings in our ms.

3. The gnomAD database is primarily composed of individuals not known to have a monogenic disease. Leptin deficiency causes quite a dramatic phenotype of hyperphagia. I anticipate they they may be under-represented in gnomAD compared to their true prevalence. (Unlike for asymptomatic or very late-onset conditions.)

Response: Again, thank you for this comment. We know that paediatric and monogenic diseases are under-represented in gnomAD. Nevertheless, the aim of our study was to estimate the prevalence of pathogenic LEP variants in the general population without known monogenic diseases. As we have stated in our discussion, it is still feasible that gnomAD encompasses multiple overweight or obese individuals causing a bias in our results. Yet, we believe that we still get a comprehensive insight in the prevalence of these putatively pathogenic variants in the general public.

---

## [Decision Letter · Decision Letter 2]

2 Sep 2022

Prevalence estimates of putatively pathogenic leptin variants in the gnomAD database

PONE-D-22-08349R2

Dear Dr. Hinney,

We’re pleased to inform you that your manuscript has been judged scientifically suitable for publication and will be formally accepted for publication once it meets all outstanding technical requirements.

Kind regards,

Alvaro Galli

Academic Editor

PLOS ONE

Additional Editor Comments (optional):

Reviewers' comments:

Reviewer's Responses to Questions

**Comments to the Author**

1. If the authors have adequately addressed your comments raised in a previous round of review and you feel that this manuscript is now acceptable for publication, you may indicate that here to bypass the “Comments to the Author” section, enter your conflict of interest statement in the “Confidential to Editor” section, and submit your "Accept" recommendation.

Reviewer #2: All comments have been addressed

Reviewer #3: (No Response)

2. Is the manuscript technically sound, and do the data support the conclusions?

Reviewer #2: Yes

Reviewer #3: Partly

3. Has the statistical analysis been performed appropriately and rigorously? 

Reviewer #2: Yes

Reviewer #3: N/A

4. Have the authors made all data underlying the findings in their manuscript fully available?

Reviewer #2: Yes

Reviewer #3: Yes

5. Is the manuscript presented in an intelligible fashion and written in standard English?

Reviewer #2: Yes

Reviewer #3: Yes

6. Review Comments to the Author

Reviewer #2: Study design is good. Methodology is appropriate. Results are in accordance with previous reported individual studies. Authors have incorporated the suggestions.

Reviewer #3: I really appreciate the authors for their effort to answer my previous comment. But I'm very sorry to say that I still can't agree with the method to estimate the pathogenicity of the variants only by in silico tools. They tried to connect the clinical data with variants from previous reports, but only partially connected. I think the prevalence calculated in this paper is over estimates.

7. PLOS authors have the option to publish the peer review history of their article (what does this mean?). If published, this will include your full peer review and any attached files.

Reviewer #2: No

Reviewer #3: No

---

## [Editor Report · Acceptance letter]

9 Sep 2022

PONE-D-22-08349R2 

Prevalence estimates of putatively pathogenic leptin variants in the gnomAD database 

Dear Dr. Hinney:

I'm pleased to inform you that your manuscript has been deemed suitable for publication in PLOS ONE. Congratulations! Your manuscript is now with our production department. 

Kind regards, 

on behalf of

Dr. Alvaro Galli 

Academic Editor

PLOS ONE